# Endocrine Disorders in a Newborn with Heterozygous Galactosemia, Down Syndrome and Complex Cardiac Malformation: Case Report

**DOI:** 10.3390/medicina59050856

**Published:** 2023-04-28

**Authors:** Ioana Rosca, Alina Turenschi, Alin Nicolescu, Andreea Teodora Constantin, Adina Maria Canciu, Alice Denisa Dica, Elvira Bratila, Ciprian Andrei Coroleuca, Leonard Nastase

**Affiliations:** 1Neonatology Department, Clinical Hospital of Obstetrics and Gynecology “Prof. Dr. P.Sirbu”, 060251 Bucharest, Romania; 2Faculty of Midwifery and Nursery, University of Medicine and Pharmacy “Carol Davila”, 020021 Bucharest, Romania; 3Emergency Clinical Hospital for Children “Grigore Alexandrescu”, 011743 Bucharest, Romania; 4Cardiology Department, Emergency Clinical Hospital for Children “M.S. Curie”, 41451 Bucharest, Romania; 5Faculty of Medicine, University of Medicine and Pharmacy “Carol Davila”, 020021 Bucharest, Romania; 6Pediatrics Department, National Institute for Mother and Child Health “Alessandrescu-Rusescu”, 020395 Bucharest, Romania; 7Pediatric Neurology Department, Clinical Psychiatric Hospital “Al. Obregia”, 041914 Bucharest, Romania; 8Obstetrics and Gynecology Department, Clinical Hospital of Obstetrics and Gynecology “Prof. Dr. P.Sirbu”, 060251 Bucharest, Romania; 9Neonatology Department, National Institute for Mother and Child Health “Alessandrescu-Rusescu”, 011061 Bucharest, Romania

**Keywords:** Down syndrome, newborn, galactosemia, hypothyroidism, hypoaldosteronism, multidisciplinary team

## Abstract

Down syndrome is the most common chromosomal abnormality diagnosed in newborn babies. Infants with Down syndrome have characteristic dysmorphic features and can have neuropsychiatric disorders, cardiovascular diseases, gastrointestinal abnormalities, eye problems, hearing loss, endocrine and hematologic disorders, and many other health issues. We present the case of a newborn with Down syndrome. The infant was a female, born at term through c-section. She was diagnosed before birth with a complex congenital malformation. In the first few days of life, the newborn was stable. In her 10th day of life, she started to show respiratory distress, persistent respiratory acidosis, and persistent severe hyponatremia, and required intubation and mechanical ventilation. Due to her rapid deterioration our team decided to do a screening for metabolic disorders. The screening was positive for heterozygous Duarte variant galactosemia. Further testing on possible metabolic and endocrinologic issues that can be associated with Down syndrome was performed, leading to hypoaldosteronism and hypothyroidism diagnoses. The case was challenging for our team because the infant also had multiple metabolic and hormonal deficiencies. Newborns with Down syndrome often require a multidisciplinary team, as besides congenital cardiac malformations they can have metabolic and hormonal deficiencies that can negatively impact their short- and long-term prognosis.

## 1. Introduction

Down syndrome is one of the most frequent chromosomal abnormalities diagnosed in newborn babies. It is also called Trisomy 21, because it is determined by the presence of an extra 21st chromosome [1,2,3,4]. A recent study [5] estimated that, in Europe in 2015, there were 419,000 people with Down syndrome. Patients affected by this syndrome can have multiple medical issues that can range from intellectual disability to congenital heart malformations, celiac disease, and endocrine disorders [3,6,7].

Studies report that the incidence of congenital cardiac malformations associated with Down syndrome is around 45–50% of the individuals affected by this syndrome [8]. The most common cardiac defects identified prenatally in fetuses with Down syndrome are atrioventricular septal defects, ventricular septal defects, secundum atrial septal defects, and persistent arterial ducts [9,10,11].

Patients with Down syndrome also have higher rates of endocrine disorders such as obesity, diabetes mellitus, short stature, vitamin D deficiency, and thyroid dysfunction [12,13,14].

Life expectancy in patients with Down syndrome has improved significantly. In United States of America in the 1950s, the median life expectancy age was 4 years; in 2010, it was as high as 58 years [15].

The medical management of newborn babies diagnosed with Down syndrome requires a multidisciplinary team. The long- and short-term prognosis of these patients is improved by reliable screening programs that help identify associated malformations as soon as possible.

## 2. Case Report

We present the case of a newborn baby, a girl, born at term. Her birth weight was 3990 g, and she was born at the gestational age of 38–39 weeks (postmenstrual gestational age). The pregnancy was surveyed by a Gynecology and Obstetrics specialist. It was considered a high-risk pregnancy, due to the facts that her mother had had a previous cesarean-section and the fetus was diagnosed in utero with a cardiac malformation. She was born through cesarean-section, and her APGAR score was 8 at one minute.

Right after birth, the newborn had a satisfactory general appearance; she had facial features characteristic for Down syndrome, no heart murmurs, and no other noticeable distress signs. Due to the fact that the medical team knew this infant had a congenital cardiac malformation, she was admitted to the Neonatal Intensive Care Unit (NICU).

The cardiology consultant diagnosed the patient with a complete atrioventricular canal defect with a large atrial septal defect (type C according to the Rastelli classification [16]), grade II left and right atrioventricular valve regurgitation, and small functional atria (Figure 1). Medical treatment with furosemide, spironolactone, and captopril was recommended and started with good tolerance and evolution. In her second day of life, a systolic murmur of grade II/VI could be heard on cardiac auscultation.

For the first seven days of life the newborn baby continued to be stable, with a good general appearance, some jaundice, and stable respiration with a SpO_2_ of 96% in atmospheric air. The neonatology medical team was able to start enteral nutrition with good digestive tolerance. She was bottle fed with milk formula (partially hydrolyzed milk formula). She had a 10% body weight loss. Blood tests, as well as abdominal and transfontanellar ultrasound, were within normal range.

In the 9th day of life, her general appearance changed; the jaundice suddenly accentuated (transcutaneous bilirubin 13.8 mg/dL), the systolic murmur accentuated (grade III/VI), and her body weight decreased. She started to require fraction of inspired oxygen (FiO_2_) of 40% to keep her SpO_2_ over 95% and she also developed signs of respiratory distress (intercostal, subcostal retractions, polypnea with approximately 70 respirations/minute, bradycardia with ventricular rate of 90–100 bpm). From this moment forward, she required continuous oxygen supplementation (both invasive and noninvasive respiratory support) as well as respiratory stimulation (kinesiotherapy).

Because of the rapid and sudden deterioration, the medical team decided to ask for a metabolic diseases panel. The results showed increased levels of galactose. Investigations continued in this direction with dosing of Galactose-1-phosphate uridylyl transferase, which confirmed the diagnosis of heterozygote galactosemia or homozygous Duarte 2 variant. The patient was started on a specific galactosemia milk formula (soy based), and 24 h later we were able to wean her off the ventilator, since her respiratory function had improved significantly.

During her entire hospitalization time, the infant had hyponatremia (dropping as low as 130 mmol/L). Therefore, the suspicion of hypoaldosteronism was raised and subsequently confirmed through aldosterone dosing. Treatment with Fludrocortisone was initiated, with a dose of 0.1 mg/day with good tolerance and favorable evolution (normalization of serum sodium level).

Screening for other possible endocrinological issues was in order at this time, and this showed hypothyroidism, thus adding to her medication Levothyroxine in a dose of 25 µg/day.

This patient spent a total of 80 days in our hospital. Her evolution was uncertain, with ups and downs, including respiratory distress syndrome of varying severity, and blood gas analysis showing high pCO_2_ values (as high as 60–70 mmHg) and low pO_2_ values. She repeatedly needed mechanical respiratory support through orotracheal intubation. Despite maximal medical treatment and the extensive efforts of a multidisciplinary medical team, she had multiple episodes of severe bradycardia and desaturation which were ultimately unresponsive to resuscitation maneuvers.

Newborn babies born with complete atrioventricular canal defect (CAVC) can present at birth with slight cyanosis due to high vascular pulmonary resistance. Cardiac insufficiency signs usually appear in the first month of life, as pulmonary neonatal vascular resistance decreases physiologically and the shunt flow increases [17,18]. In our patient, cardiac failure signs were early, at one week postnatal age. The newborn had tachypnea, needed oxygen supplementation, and had feeding difficulties and growth failure, making her congenital cardiac malformation the main cause for her respiratory distress.

Another cause for respiratory distress in newborn babies is infection. Patients with cardiac failure can have recurrent respiratory infections [17,18]. Ventilation pneumonia is a common occurrence in infants that require intubation and mechanical ventilation for a large amount of time. It is usually considered a nosocomial infection and definitive diagnosis criteria include: more than 48 h of mechanical ventilation; changes in blood gases; a need to increase ventilation parameters; temperature instability; tachypnea, wheezing, cough, or abundant secretions; changes in cardiac rhythm; and leukocytosis. Three signs or symptoms from those listed above are necessary to diagnose ventilation pneumonia [19]. Our patient had clinical criteria for ventilation pneumonia and needed prolonged mechanical ventilation.

In patients with CAVC and Down syndrome, pulmonary hypertension appears earlier and has increased severity due to genetic factors, pulmonary hypoplasia, and associated chronic hypoventilation [17,20]. Among the causes of pulmonary hypertension in newborn babies, there is alveolar hypoxia due to respiratory failure, alveolar hypoventilation due to abnormalities of the central nervous system, and acidosis and shock due to left ventricular dysfunction [18]. Our patient met these criteria. The X-ray (Figure 2) had non-specific findings: cardiac enlargement with signs of overload on pulmonary circulation and dilatation of the pulmonary artery [17]. In our case, the X-ray showed these suggestive findings.

We took into consideration other causes for respiratory failure—narrow respiratory airway, laryngomalacia, hypotonia—that are specific to a patient with Down syndrome. Prolonged endotracheal intubation can be associated with subglottic stenosis, tracheobronchomalacia, and chronic lung disease. We did take all these possible diagnoses into consideration for our patient. The ENT specialist was able to exclude laryngomalacia and congenital obstruction of upper respiratory airways.

## 3. Discussion

Down syndrome, besides being the most frequently identified chromosomal abnormality, is also one of the most studied in the last 150 years. The most frequent malformations associated with Down syndrome are cardiac, gastrointestinal, musculoskeletal, urinary, and endocrine [7,11,21,22]. In the presented case, the congenital cardiac malformation was diagnosed before birth, which required the patient to be taken in charge in a level III maternity ward with an advanced NICU and a multidisciplinary team.

As we previously mentioned, this patient presented with respiratory distress almost throughout her entire hospitalization. This led the medical team to multiple differential diagnoses, such as pneumonia, pulmonary edema, and ear, nose and throat (ENT) pathology. Confirming the cardiac malformation in the first 24 h of life allowed the medical team to initiate specific treatment as soon as possible and to initiate advanced monitoring. There were multiple causes for her respiratory distress, starting with her complex cardiac malformation, pulmonary hypertension secondary to the cardiac malformation, a narrow upper respiratory tract that is specific to Down syndrome, gastrointestinal reflux, and her metabolic and endocrine associated disorders (hypoaldosteronism, hypothyroidism, galactosemia).

Galactosemia is an inborn error of metabolism that affects carbohydrate metabolism. Severe galactose-1-phosphate uridylyl transferase (GALT) deficiency and classic galactosemia can be deadly for newborns [23,24,25]. There are certain variants of galactosemia. The Duarte variant is characterized by GALT activity of about 50% in the red blood cells (if homozygotes). Heterozygotes for the Duarte variant have about 75% GALT activity [26]. Although classic galactosemia is considered to be a medical emergency, the Duarte variant is considered to be asymptomatic and usually does not necessitate a galactose restricted diet [27,28]. Our patient had no digestive symptoms specific to classic galactosemia (she had good digestive tolerance, no loose stools, no emesis). Her blood work did not show metabolic acidosis but respiratory acidosis, which is atypical for galactosemia. However, the respiratory distress and respiratory acidosis improved significantly when a galactose-restrictive diet with a soy-based formula was initiated.

Hypoaldosteronism is an endocrine disorder characterized by aldosterone deficiency or defective aldosterone activity on the tissue level. The severity is usually inversely related to age [29]. The most frequent clinical findings are hyperkaliemia, hyponatremia associated with hypovolemia, and metabolic acidosis [30]. In our patient, what raised the suspicion was persistent hyponatremia, despite appropriate intravenous correction. We decided to verify the serum cortisol level (which was within normal range) and serum aldosterone level (low level). Thus, oral treatment with fludrocortisone was initiated.

When compared to the general population, patients with Down syndrome are 25–38-fold more likely to have thyroid dysfunction [31,32]. In patients with Down syndrome, screening for thyroid using thyroid-stimulating hormone (TSH), free T4, and free T4 should be started at birth and repeated at 6 months, 12 months, and annually thereafter [33,34,35]. This recommendation is supported by the fact that, in our patient, the initial newborn screening was negative for hypothyroidism. The medical team decided to repeat thyroid functional tests and could diagnose hypothyroidism and initiate appropriate treatment with Levothyroxine.

The decision to start feeding our patient with hydrolyzed milk was made for the prevention of necrotizing enterocolitis (NEC), as the newborn was diagnosed with severe heart disease. NEC is an ischemic and inflammatory necrosis of the bowel, primarily affecting premature neonates after the initiation of enteral feeding, but 10% of all cases of NEC occur in term infants [36]. Risk factors for this group of term neonates include congenital heart disease with presumed low intestinal perfusion [20]. The pathogenesis of NEC is multifactorial, and additional risk factors for this pathology include ischemia, intestinal dysbiosis, and formula feedings; almost all newborns with NEC received enteral nutrition before the onset of the disease [20,36]. Infants with congenital heart disease may have compromised bowel perfusion, making them susceptible to ischemic injury; hypoxic/ischemic events play a greater role in the pathogenesis of NEC [36]. For the prevention of NEC, several strategies are needed, including the exclusive use of breast milk and minimizing exposure to empiric antibiotic therapy [20]. Enteral feeding provides a necessary substrate for the proliferation of enteric pathogens. Hyperosmolar formulas or medications may alter mucosal permeability and cause mucosal damage. Human milk, with the benefit of providing immunoprotective as well as local growth promoting factors, significantly lowers the risk of NEC [36,37]. Moreover, enteral immunoglobulin A-immunoglobulin G (IgA-IgG) feeding also decreased the risk for NEC in preliminary clinical studies [36]. Our patient could not be fed with breast milk because the mothers’ lactate secretion was installed late after the cesarean section; she initially had hypogalactia and later was discharged from our hospital unit. In our unit, there is no breast milk bank, and in this situation, we chose to feed the newborn with hydrolyzed milk, the most appropriate formula in this case with risk factors for NEC. Another risk factor for NEC is bacterial colonization; bacteria including Escherichia coli, Klebsiella species, Clostridia species, and Staphylococcus epidermidis are implicated in NEC [36].

This case is unique because of the coexistent conditions of our patient. In patients with Down syndrome, congenital cardiac malformations are quite frequent, ranging from 20 to 57.9% depending on the study [38]. Most studies report atrioventricular septal defect as being one of the most frequent congenital heart diseases associated with Down syndrome [39,40,41]. Although our patient had one of the most frequent congenital heart defects associated with Down syndrome, it was not isolated, being associated with other defects, thus being life threatening and requiring medical treatment. Hypothyroidism is known to be frequently associated with Down syndrome [42]. We did not find reports of other cases of Down syndrome associated with galactosemia, but there are recent studies and theories about different metabolic deficiencies associated with Down syndrome [43]. Although not unexpected, the association of Down syndrome, congenital heart disease, and hypothyroidism, with the addition of Duarte galactosemia, is of interest due to the challenges multiple-pathology cases imply for the medical team. The sudden deterioration of this patient had multiple etiological factors: pulmonary, cardiac, neurological, metabolic, and hormonal, and the management was complex.

Communication with the parents of the child with Down syndrome is very important in the diagnosis, treatment, and follow-up of such patients. Their families have complex needs and lives. It is important for them to understand the importance of a multidisciplinary team and have a good collaboration with such a team [44].

Limitations in our approach to this case arises from the fact that all interdisciplinary consultations occurred in other medical units (cardiology, ENT, endocrinology, pediatric neurology, genetics) and required the newborn baby being transported to those hospitals, involving additional stress for the patient.

Although not unexpected, the association of Down syndrome, congenital heart disease, and hypothyroidism, with the addition of Duarte galactosemia, is of interest due to the challenges multiple-pathology cases imply for the medical team. The sudden deterioration of this patient had multiple etiological factors: pulmonary, cardiac, neurological, metabolic, and hormonal, and the management was complex.

In the medical field, maybe more than anywhere else, it is important to keep an open mind and think outside of the box when searching for the right diagnosis and treatment.

Even if there are known associations, such as Down syndrome with congenital cardiac malformations and hypothyroidism, we should always take into consideration other possible diagnoses.

Any newborn baby can have congenital metabolic disease that are clinically difficult to diagnose.

The newborn screening national program should include as many metabolic disorders as possible.

## 4. Conclusions

In conclusion, the case we presented emphasizes the need for a multidisciplinary team in the management of such a complex case. Endocrine dysfunction had an important impact on the evolution of this patient, thus adding to the importance of screening for endocrine disorders in patients with Down syndrome, even if the initial evaluation is normal.

Regardless of the severity of the disease and the prognosis, any newborn has the right to advanced medical assistance. The parents of this newborn were aware of the severity of the disease before she was born, decided to continue with the pregnancy, and did not regret their decision as far as we know.

## Figures and Tables

**Figure 1 medicina-59-00856-f001:**
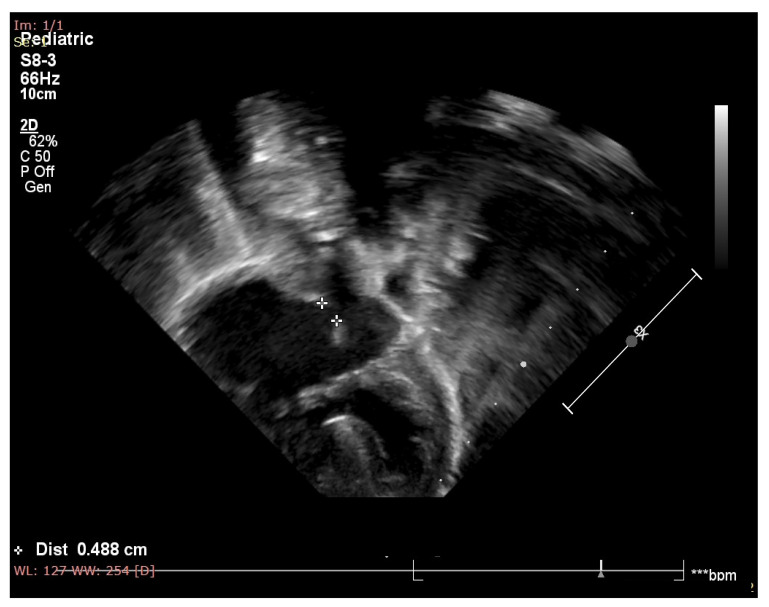
Echocardiography of the patient showing atrial septal defect, inlet type ventricular septal defect, one atrioventricular valve and balanced ventricles.

**Figure 2 medicina-59-00856-f002:**
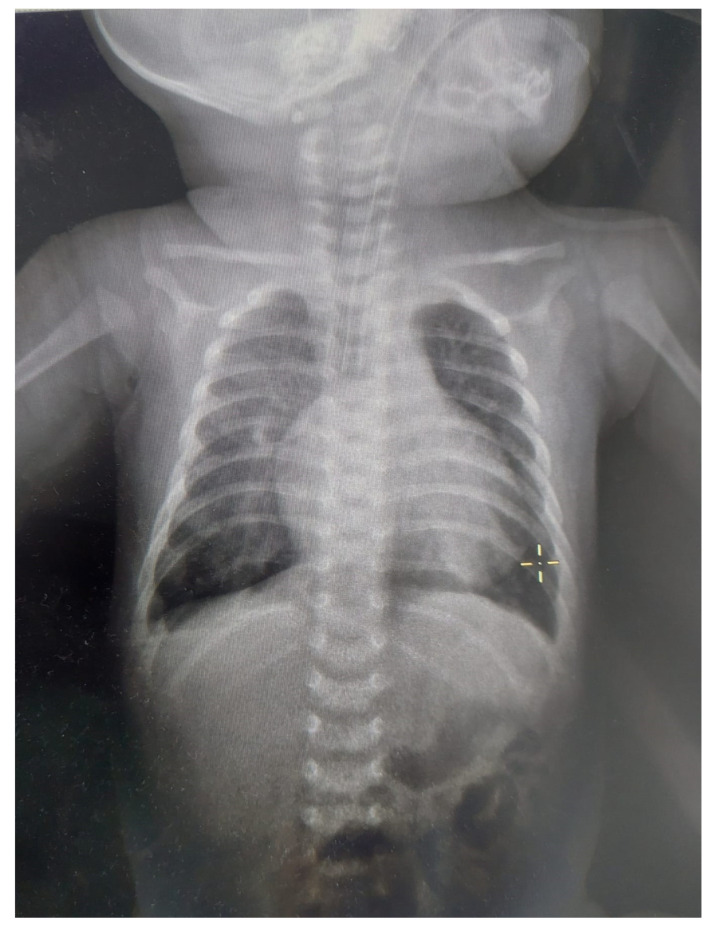
X-ray for the patient at approximately one month age. The patient was intubated and mechanically ventilated. We noticed the enlarged cardio-thoracic index (>0.6), pulmonary hypoventilation, and accentuation of pulmonary hilum.

## Data Availability

Data sharing not applicable.

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
