# Peer review of "Endocrine Disorders in a Newborn with Heterozygous Galactosemia, Down Syndrome and Complex Cardiac Malformation: Case Report"

_medicina, 2023, doi:10.3390/medicina59050856_

Round 1
Reviewer 1 Report
General comment
General comment
Newborns with Down syndrome often require a multidisciplinary team as besides congenital cardiac malformations they can have metabolic and hormonal deficiencies that can negatively impact their short- and long-term prognosis. The authors present the case of a newborn with Down Syndrome.
Specific points
1- I think, the following sentence need re-editing in the introduction section
-Down syndrome is the most frequent chromosomal abnormality diagnose in new born babies.
2- I think, the authors should write “transfontanellar” instead of “transfontanelar” in case report section. I think, the authors should write “within normal range” instead of “withing normal range” in case report section
-Blood tests as well as abdominal and transfontanelar ultrasound were withing normal range
3- I think, the authors should not use number at the beginning of the sentence. e.g. Seventy respirations/minute………….
-70 respirations/minute, bradycardia with 90 ventricular rate of 90-100 bpm).
4-I think, the authors should write “dysfunction” instead of “disfunction” in discussion section. Both “dysfunction” and “disfunction” are correct but dysfunction is the generally accepted preferred spelling that most people use.
-When compared to the general population, patients with Down syndrome are 25-38 fold more likely to have thyroid disfunction [27,28].
Author Response
Thank you for agreeing to review our article. It is a real honor to have your opinion so that we can improve the manuscript and highlight the complexity of the case.
I have added all the proposed changes to the manuscript.

Reviewer 2 Report
1-What was the reason for starting feeding with hydrolyzed milk?
2-Briefly summarize why this case is unique with medical literature references.
3-talk more about Diagnostic reasoning including differential diagnosis.
4-What are one or more “take-away” lessons?
5-what is Strengths and limitations in your approach to this case.
6-Is the informed consent form completed by the parents?
Author Response
1-What was the reason for starting feeding with hydrolyzed milk?
The decision to start feeding our patient with hydrolyzed milk was prevention for necrotizing enterocolitis (NEC) as the newborn was diagnosed with severe heart disease. NEC is an ischemic and inflammatory necrosis of the bowel primarily affecting premature neonates after the initiation of enteral feeding, but 10% of all cases of NEC occur in term infants [1]. Risk factors for this group of term neonates include congenital heart disease with presumed low intestinal perfusion [2]. The pathogenesis of NEC is multifactorial and additional risk factors for this pathology include ischemia, intestinal dysbiosis and formula feedings, almost all newborns with NEC received enteral nutrition before the onset of the disease [1,2]. Infants with congenital heart disease may have compromised bowel perfusion, making them susceptible to ischemic injury, hypoxic/ischemic events play a greater role in pathogenesis of NEC [1]. For the prevention of NEC, several strategies are needed, including the exclusive use of breast milk and minimizing exposure to empiric antibiotic therapy [2]. Enteral feeding provides necessary substrate for proliferation of enteric pathogens. Hyperosmolar formulas or medications may alter mucosal permeability and cause mucosal damage. Human milk, with the benefit of providing immunoprotective as well a local growth promoting factors, significantly lowers the risk for NEC [1,3]. Moreover, enteral immunoglobulin A-immunoglobulin G (IgA-IgG) feeds also decreased the risk for NEC in preliminary clinical studies [1]. Our patient could not be fed with breast milk because the mothers’ lactate secretion was installed late after the cesarean section, she was initially hypogalactic and later discharged from our hospital unit. In our unit there is no breast milk bank and, in this situation, we chose to feed the newborn with hydrolyzed milk, the most appropriate formula in this case with risk factors for NEC. Another risk factor for NEC is bacterial colonization, including Escherichia coli, Klebsiella species, Clostridia species, and Staphylococcus epidermidis are implicated in NEC [1].
2-Briefly summarize why this case is unique with medical literature references.
This case is unique because of the coexistent conditions of our patient. In patients with Down Syndrome, congenital cardiac malformations are quite frequent ranging from 20 to 57.9% depending on the study [4]. Most studies report atrioventricular septal defect as being one of the most frequent congenital heart diseases associated with Down syndrome [5–7]. Although our patient had one of the most frequent congenital heart defects associated with down syndrome it was not isolated, being associated with other defects, thus being life threatening and requiring medical treatment. Hypothyroidism is known to be frequently associated with Down syndrome, with a prevalence 28 to 38 times higher than the general population [8]. We did not find reports of other cases of Down syndrome associated with galactosemia but there are recent studies and theories about different metabolic deficiencies associated with Down syndrome [9]. Although not unexpected, the association of Down Syndrome, congenital heart disease and hypothyroidism, adding Duarte galactosemia, is of interest due to the challenges multiple-pathology cases imply for the medical team. The sudden deterioration of this patient had multiple etiological factors: pulmonary, cardiac, neurological, metabolic, and hormonal and the management was complex.
3-talk more about Diagnostic reasoning including differential diagnosis.
Newborn babies born with complete atrioventricular canal defect (CAVC) can present at birth with slight cyanosis due to high vascular pulmonary resistance. Cardiac insufficiency signs usually appear in the first month of life as pulmonary neonatal vascular resistance decreases physiologically and the shunt flow increases [10,11]. In our patient, cardiac failure signs were early, at one week postnatal age. The newborn had tachypnea, needed oxygen supplementation, had feeding difficulties and growth failure, making her congenital cardiac malformation the main cause for her respiratory distress.
Another cause for respiratory distress in newborn babies is infection. Patients with cardiac failure can have recurrent respiratory infections [10,11]. Ventilation pneumonia is a common occurrence in infants that require intubation and mechanical ventilation for large amount of time. It is usually considered a nosocomial infection and definitive diagnosis criteria include: more than 48h of mechanical ventilation; changes in blood gases; need to increase ventilation parameters; temperature instability; tachypnea, wheezing, cough, abundant secretions; changes in cardiac rhythm; leukocytosis. Three signs or symptoms from the ones listed above are necessary to diagnose ventilation pneumonia [12]. Our patient had clinical criteria for ventilation pneumonia and needed prolonged mechanical ventilation.
In patients with CAVC and Down Syndrome, pulmonary hypertension appears earlier and has increased severity due to genetic factors, pulmonary hypoplasia and chronic hypoventilation associated [3,10]. Among the causes of pulmonary hypertension in newborn babies there is alveolar hypoxia due to respiratory failure, alveolar hypoventilation due to abnormalities of the central nervous system, acidosis and shock due to left ventricular disfunction [11]. Our patient meets these criteria. The X-ray has non-specific findings: cardiac enlargement with signs of overload on pulmonary circulation and dilatation of the pulmonary artery [10]. In our case, the X-ray shows these suggestive findings.
We did take into consideration other causes for respiratory failure: narrow respiratory airway, laryngomalacia, hypotonia that are specific to a patient with Down Syndrome. Prolonged endotracheal intubation can be associated with subglottic stenosis, tracheobronchomalacia and chronic lung disease. We did take all these possible diagnosis in consideration for our patient. The ENT specialist was able to exclude laryngomalacia and congenital obstruction of upper respiratory airways.
4-What are one or more “take-away” lessons?
In the medical field, maybe more that anywhere else it is important to keep an open mind and think out of the box when searching for the right diagnosis and treatment.
Even if there are known associations such as Down syndrome with congenital cardiac malformations and hypothyroidism, we should always take into consideration other possible diagnosis,
Any newborn baby can have congenital metabolic disease that are clinically difficult to diagnose.
Newborn screening national program should include as many metabolic disorders as possible.
5-what is Strengths and limitations in your approach to this case.
A strength in our approach to this case could be considered we did not limit ourselves to usual causes for respiratory failure and we were able to identify a possible rare cause such as galactosemia through genetic testing.
Limitations in our approach to this case arises from the fact that all interdisciplinary consults were done in other medical units (cardiology, ENT, endocrinology, pediatric neurology, genetics) and required the newborn baby being transported to those hospitals, involving additional stress for the patient.
6-Is the informed consent form completed by the parents?
Yes
References
- Gomella, T.L.; Cunningham, M.D.; Eyal, F.G.; Tuttle, D. Neonatology : Management, Procedures, on-Call Problems, Diseases, and Drugs; Fried, A.K., Lebowitz, H., Eds.; 7th ed.; McGraw-Hill: New York, 2013; ISBN 9780071768016, 9780071816991, 0071768017, 0071816992.
- Eichenwald, E. c; Hansen, R.A.; Martin, C.R.; Stark, A.R. Cloherty and Stark’s Manual of Neonatal Care; 2021; Vol. 8 Ed; ISBN 9788194864554.
- Maria, S. Aspecte Practice in Nutritia Neonatala; 1st ed.; Editura Universitară „Carol Davila”, 2013;
- Santoro, S.L.; Steffensen, E.H. Congenital Heart Disease in Down Syndrome – A Review of Temporal Changes. J. Congenit. Cardiol. 2021, 5, 1–14, doi:10.1186/s40949-020-00055-7.
- Ujuanbi Amenawon, S.; Onyeka Adaeze, C. Prevalence and Pattern of Congenital Heart Disease among Children with Down Syndrome Seen in a Federal Medical Centre in the Niger Delta Region, Nigeria. J. Cardiol. Cardiovasc. Med. 2022, 7, 030–035, doi:10.29328/journal.jccm.1001129.
- Ko, J.M. Genetic Syndromes Associated with Congenital Heart Disease. Korean Circ. J. 2015, 45, 357–361, doi:10.4070/kcj.2015.45.5.357.
- Benhaourech, S.; Drighil, A.; Hammiri, A. El Congenital Heart Disease and Down Syndrome: Various Aspects of a Confirmed Association. Cardiovasc. J. Afr. 2016, 27, 287–290, doi:10.5830/CVJA-2016-019.
- Amr, N.H. Thyroid Disorders in Subjects with Down Syndrome: An Update. Acta Biomed. 2018, 89, 132–139, doi:10.23750/abm.v89i1.7120.
- Caracausi, M.; Ghini, V.; Locatelli, C.; Mericio, M.; Piovesan, A.; Antonaros, F.; Pelleri, M.C.; Vitale, L.; Vacca, R.A.; Bedetti, F.; et al. Plasma and Urinary Metabolomic Profiles of Down Syndrome Correlate with Alteration of Mitochondrial Metabolism. Sci. Rep. 2018, 8, 1–16, doi:10.1038/s41598-018-20834-y.
- Nicolescu, A.; Cinteză, E. EsenÈ›ialul În Cardiologia Pediatruca; Amaltea, 2022; ISBN 9789731622255.
- Maria, S.; Andreea-Luciana, A. Urgențe Neonatale; Tehnopress, 2018; ISBN 978-606-687-363-5.
- Florea, I. Tratat de Pediatrie; ALL: Bucharest, 2019; ISBN 978-606-587-550-0.

Round 2
Reviewer 2 Report
Hello and thank you for your reply
What you have mentioned about NEC is correct. Although the introduction of hydrolyzed formula does not affect the results of your study, it has no effect on the prevention of this disease, and its inappropriate use can have long-term complications at the time of the initiation of infant feeding at the age of 6 months.
The association of galactosemia with Down syndrome can be an incidental finding, unlike the other co-morbidities you have mentioned. Therefore, no conclusion can be made about the necessity of investigating galactosemia in patients with Down syndrome.
Author Response
The presented case highlights the multitude of diseases associated with Down Syndrome, some well-known such as cardiac malformations, and others less known such as endocrine and metabolic ones.
The diagnosis of galactosemia in the presented case changed the patient's prognosis, improving the general condition and allowing rapid extubation after changing the milk formula.
Our team does not consider it necessary to screen all newborns with Down syndrome for galatosemia, but it is known that they have an increased risk for metabolic diseases. In our country, metabolic screening is not carried out at birth, but it could be useful in complex cases, such as the one reported.
